# Field Evaluation of Cypermethrin, Imidacloprid, Teflubenzuron and Emamectin Benzoate against Pests of Quinoa (*Chenopodium quinoa* Willd.) and Their Side Effects on Non-Target Species

**DOI:** 10.3390/plants10091788

**Published:** 2021-08-27

**Authors:** Luis Cruces, Eduardo de la Peña, Patrick De Clercq

**Affiliations:** 1Department of Entomology, Faculty of Agronomy, Universidad Nacional Agraria La Molina, Lima 12-056, Peru; 2Department of Plants & Crops, Faculty of Bioscience Engineering, Ghent University, B-9000 Ghent, Belgium; eduardo.deLaPena@ugent.be (E.d.l.P.); patrick.declercq@ugent.be (P.D.C.); 3Instituto de Hortofruticultura Subtropical y Mediterránea “La Mayora (IHSM-UMA-CSIC), Estación Experimental “La Mayora”, 29750 Malaga, Spain

**Keywords:** insecticides, quinoa pests, side effects, natural enemy, IPM

## Abstract

During the last few years, quinoa, a traditional Andean crop, has been cultivated at low elevations where pest pressure is high and farmers resort to intensive use of insecticides. This field study investigated the impact of four insecticides (cypermethrin, imidacloprid, teflubenzuron and emamectin benzoate) on insect pests of quinoa and their side effects on the arthropod community at the coastal level of Peru, by analysing the species composition, species diversity and population density. The arthropod community was examined with pitfall traps (for ground dwelling species), plant samplings (for pests and their natural enemies that inhabit the crop), and yellow pan traps (to catch flying insects). The results demonstrated that *Macrosiphum euphorbiae*, *Frankliniella occidentalis* and *Spoladea recurvalis* were efficiently controlled by cypermethrin and imidacloprid; the latter compound also showed long-term effects on *Nysius simulans*. Teflubenzuron and emamectin benzoate proved to be efficient to control *S. recurvalis*. Imidacloprid had the strongest adverse effects on the arthropod community in terms of species diversity, species composition and natural enemy density as compared to the other insecticides. Findings of this study may assist farmers intending to grow quinoa at the coastal level in selecting the most appropriate insecticides under an integrated pest management approach.

## 1. Introduction

Quinoa is an Andean grain that has increasingly gained international interest due to its high nutritional value [1]. During the last decade, the cultivated area of quinoa has substantially increased in South American countries such as Ecuador, Chile and particularly Bolivia and Peru [2,3,4]. In the case of Peru, the production areas have extended to lower elevations, reaching even the coastal level [5]. Moreover, there have been attempts of cultivating quinoa outside of its Andean place of origin, in countries such as the United States of America, the United Kingdom and a number of other European countries [6,7]. Therefore, recent efforts have been made to adapt the cultivation of quinoa under non-Andean conditions, including studies regarding to the entomofauna associated with the crop [5,8,9,10,11,12]. 

In the Andes, at an altitude between 2300 and 3800 m a.s.l., quinoa is traditionally cultivated since ancient times [7]. In this region *Eurysacca melanocampta* Meyrick and *Eurysacca quinoae* Povolvý are the key pests of quinoa, causing damage by feeding on the developing grains; a range of other phytophagous insects are considered of minor importance [2,6].

At the coastal level, the number of relevant phytophagous insects infesting quinoa is substantially larger. These include species of wide distribution such as the cosmopolitan aphids (*Macrosiphum euphorbiae* (Thomas)), thrips (*Frankliniella occidentalis* (Pergande)) and leafminer flies (*Liriomyza huidobrensis* Blanchard); and also others of neotropical distribution such as certain true bugs (*Nysius simulans* Stål and *Liorhyssus hyalinus* (Fabricius)) and lepidopteran larvae (i.e., *E. melanocampta*, *Chloridea virescens* (Fabricius), *Spoladea recurvalis* Fabricius)) that feed on the developing grains [2,5,11]. Under this scenario, farmers may be prompted to apply more pesticides than quinoa growers from the highlands. Hence the need for exploring a range of chemical compounds that may be suitable for use in an integrated pest management (IPM) programme. As in other field crops [13], selective insecticides with a more favourable toxicological profile to the natural enemy community may be a valuable tool for IPM in quinoa.

Cypermethrin is commonly used by the quinoa growers [6]. This insecticide of the pyrethroid group is a nonpersistent sodium channel modulator, characterized by a broad spectrum activity. The compound acts by direct contact, causing neuronal hyperexcitation alongside the axon [10,14]. Due to its relatively short residual effects and lower price, this pesticide is often overused, causing environmental issues and promoting resistance in pest insects [5,10]. The adverse effects of cypermethrin on non-target organisms are widely documented [15,16].

Imidacloprid is one of the most widely used insecticides worldwide [17,18,19] and it is also commonly used by farmers in coastal areas of quinoa production [20]. This compound may effectively control a range of phytophagous insects noted to be pests of quinoa (including aphids, thrips, true bugs and some lepidopteran species). This neurotoxic insecticide of the neonicotinoid group, is an acetylcholine receptor agonist with broad spectrum and highly systemic activity, acting by ingestion and direct contact, causing neuronal hyperexcitation at the level of the synapses [5,21]. Toxicity of imidacloprid to non-target organisms, including beneficial species such as pollinators (i.e. bees) and natural enemies has been documented in different crops, but there are presently no reports for quinoa [15,16,22,23,24,25,26,27]. Due to these adverse effects, particularly to the bees, this active ingredient has been banned in Europe [28]. 

Teflubenzuron has a more favourable environmental profile, with lesser toxicity to a range of non-target organisms as compared to the broad-spectrum compounds, and thus may be considered as a tool for an IPM program in quinoa [29]. This insect growth regulator (IGR) of the benzoylphenylureas group is a highly active inhibitor of chitin synthesis, aimed mainly at lepidopteran larvae. This compound is considered to be safer to the beneficial fauna (especially in the adult stage) than pyrethroids and neonicotinoids, although there are reports of its toxicity towards a number of arthropod predators [16,30,31,32]. 

Emamectin benzoate, a neurotoxic insecticide of the avermectin group, is another insecticide reported to be more selective against lepidopteran larvae. Although toxicity to some natural enemies and non-target arthropods have been reported, this insecticide is considered less harmful to beneficial arthropods as compared with broad spectrum compounds [16,33]. The insecticide acts mainly by ingestion causing paralysis in the insect by activating allosterically the glutamate-gated chloride channels in the synapses [16,30,34,35,36]. 

The aim of the present field study was to examine the effects of two broad spectrum insecticides (cypermethrin and imidacloprid) and two selective insecticides (teflubenzuron and emamectin benzoate), with different modes of action and from different chemical groups, against quinoa pests in Peru and record their side effects on non-target arthropods by analysing the species composition, species diversity and population density in quinoa fields at the coastal level (a region with potential areas for quinoa production). To assess the effects of these insecticides on the arthropod community, including phytophagous and beneficial species, we combined three sampling techniques (i.e., pitfall traps, plant sampling and yellow pan traps) targeting groups from different ecological habitats. The findings of this field study should be of special interest to quinoa farmers and agricultural extensionists from Andean and non-Andean countries who are exploring the cultivation of quinoa, to use insecticides of higher compatibility with natural biological control as part of an IPM approach.

## 2. Results

### 2.1. Effects on the Composition of the Arthropod Fauna

The NMDS-plots show the distances between treatments concerning the composition of the arthropods collected from 24 September 2017 to 11 December 2017, with the different sampling methodologies (in Figure 1), based on the Bray–Curtis dissimilarity index. Ellipses are formed by the replications of each treatment (based on the presence and abundance of species) and the closeness or distances between them reflect their similarities and dissimilarities, respectively. Some morphospecies are shown in the NMDS-plots to depict the differences in the species composition collected in each insecticide treatment: the distance of a morphospecies to the centre of an ellipse (treatment) reflects its scarcity or even absence in the treatment.

As to the pitfall trap data, the proximity of the ellipses in the NMDS-plot (Figure 1a) suggests high similarity among the treatments in terms of the composition of ground dwelling species, which was confirmed by the PerMANOVA test that indicated no significant differences between treatments (F–model_4,10_ = 0.7537, *p* = 0.819). Likewise, for the flying insects collected at the top of the canopy with the pan traps (Figure 1c), no significant differences between treatments were found (F–model_4,10_ = 1.066, *p* = 0.441). For the plant sampling data (i.e., the specimens collected from the quinoa plants), however, the ellipse corresponding to the imidacloprid treatment in the NMDS-plot (Figure 1b) is separated from the others, suggesting high dissimilarity between imidacloprid and the other treatments; in this case, the test was significant (F–model_4,10_ = 2.835, *p* = 0.001).

### 2.2. Effects on Diversity of Arthropods

#### 2.2.1. Structure

Rank abundance curves of the morphospecies collected in the period between 24 September 2017 and 11 December 2017, were calculated for each insecticide treatment (Figure 2). For the pitfall trap data, the corresponding curves for the treatments and the untreated control have similar patterns, except for imidacloprid in which a slightly more pronounced slope can be observed (Figure 2a), indicating that the imidacloprid treatment affected the evenness of the ground dwelling arthropod community to a higher degree than the other insecticides. When applying the ANOVA, significant differences between treatments were found for the Shannon (F_4,8_ = 4.109, *p* = 0.042) and Simpson’s dominance (F_4,8_ = 4.038, *p* = 0.038) indices. The Duncan test confirmed that imidacloprid had a significantly greater impact on the species equitability (with lowest value of Shannon index), preventing dominance of certain taxa (with the lowest value of Simpson’s dominance index) (Table 1). 

As to the specimens collected from the quinoa plants, the corresponding curve for imidacloprid markedly differs from the other treatments and the untreated control, due to the lower number of species collected (Figure 2b). However, no significant differences between treatments for the Shannon (F_4,8_ = 2.57, *p* = 0.119) and Simpson’s dominance (F_4,8_ = 1.81, *p* = 0.220) indices were found (Table 1). For the pan trap data, the curves for all treatments and the control have similar patterns (Figure 2c), suggesting a similar distribution of flying species in the community over all plots. The ANOVA confirmed no significant differences between treatments in terms of the Shannon (F_4,8_ = 0.34, *p* = 0.841) and Simpson’s dominance (F_4,8_ = 0.45, *p* = 0.771) indices.

#### 2.2.2. Species Richness

Significant differences between treatments were found in species richness of the ground dwelling arthropods, measured by the Margalef index (F_4,8_ = 5.55, *p* = 0.019). The lowest species richness was obtained with imidacloprid, being significantly inferior to that of the teflubenzuron and emamectin benzoate treatments and the untreated control, but without significant differences with cypermethrin. Significant differences were also found for the insects collected from the plants (F_4,8_ = 4.76, *p* = 0.029), with the imidacloprid treatment having the lowest value. For the pan trapping data, no significant differences in species richness of flying insects between treatments and the control were found (F_4,8_ = 0.49, *p* = 0.747) (Table 1).

### 2.3. Effects on Functional Species Pools

#### 2.3.1. Phytophagous Group

Four herbivorous species infested the plots in relatively high abundance: *Spoladea recurvalis,* which appeared at the early stages of the crop phenology and *Macrosiphum euphorbiae, Frankliniella occidentalis* and *Nysius simulans*, the incidence of which in all plots was recurrent throughout the cropping season. The mean numbers of these species per plant were compared between treatments (Table 2).

The statistical analysis indicated that all insecticides were efficient to reduce *S. recurvalis* incidence after the first application as compared to the untreated control (F_4,8_ = 7.73, *p* = 0.007). Since this pest had disappeared in the treated plots, the effects after the second application could not be evaluated, neither at day 6 nor at day 69 after the application.

Significant differences in the numbers of *M. euphorbiae* were observed, 6 days after the first (F_4,8_ = 28.73, *p* < 0.001) and 6 days after the second applications (F_4,8_ = 7.32, *p* = 0.008), and also 69 days after the second application (F_4,8_ = 7.80, *p* = 0.007). Six days after the first application, the lowest mean number per plant was obtained with imidacloprid, differing significantly from the numbers observed in the teflubenzuron and emamectin benzoate treatments and in the untreated control, whereas the imidacloprid and cypermethrin treatments were similar. On day 6 after the second application, significantly lower aphid numbers were registered for the imidacloprid and cypermethrin treatments, whereas the teflubenzuron and emamectin benzoate treatments were similar to the untreated control. At the last sampling, 69 days after the second application, the aphid numbers were similar in the teflubenzuron, cypermethrin and the untreated plots; the highest aphid abundance was recorded with emamectin benzoate and the lowest with imidacloprid. 

Significant differences in *F. occidentalis* numbers were observed 6 days after the first (F_4,8_ = 9.47, *p* = 0.004) and 6 days after second applications (F_4,8_ =171.17, *p* < 0.001), and 69 days after the second application (F_4,8_ = 46.76, *p* < 0.001). On day 6 after the first application, the lowest mean values were observed with imidacloprid and cypermethrin, and no significant differences were found between teflubenzuron, emamectin benzoate and the untreated control. Six days after the second application, the lowest mean values were also obtained with imidacloprid and cypermethrin, and lower thrips numbers were found in the teflubenzuron and emamectin benzoate plots as compared to the untreated control. At the last sampling date, 69 days after the second application, the lowest thrips numbers were obtained with imidacloprid, whereas the cypermethrin and teflubenzuron treatments and the untreated control had similar numbers; the thrips numbers in the emamectin benzoate plots were significantly higher than in the other plots.

Significant differences in *N. simulans* numbers were not seen until 69 days after the second application (F_4,8_ = 25.87, *p* < 0.001). The lowest mean values were obtained with imidacloprid and cypermethrin (in this order); the mean value obtained in the untreated control was similar to that in the cypermethrin and emamectin benzoate treatments, whereas the highest value was recorded in the teflubenzuron treatment.

The phytophagous insects were also examined in the pitfall and pan traps. The mean cumulative number of *N. simulans* and *F. occidentalis,* recorded from 24 September 2017 to 11 December 2017, were compared between treatments (Table 3). No significant differences between treatments were found as to *N. simulans* numbers trapped with pitfall traps (F_4,8_ = 1.62, *p* = 0.261) nor as to those collected with pan traps (F_4,8_ = 0.42, *p* = 0.792). 

There were significant differences in *F. occidentalis* numbers collected with pan traps (F_4,8_ = 11.18, *p* = 0.002). The lowest mean values were observed in the cypermethrin treatments, whereas the highest thrips numbers were noted in the teflubenzuron plots; the emamectin benzoate, imidacloprid and untreated plots had similar values. Adults of *L. huidobrensis* were also collected in relatively high abundance in the pan traps, but as they appeared at the later stages of the crop phenology, leaf miner larvae were not observed in the examined leaves and therefore they were not considered in the analysis.

#### 2.3.2. Natural Enemies

The most recurrent natural enemy groups found on the collected plants were Aphidiinae wasps (adult and parasitized aphids), predatory true bugs (*Metacanthus tenellus* Stål, *Rhinacloa* sp., and *Nabis capsiformis* Germar), syrphid larvae (*Allograpta* sp.) and chrysopid larvae. The mean cumulative numbers of individuals per plot of these groups were compared between treatments (Table 4). Other predators such as coccinellids and hemerobiids were also found in some plots, but their incidence was irregular throughout the monitoring and in small numbers; therefore, they were excluded from the analysis. 

Numbers of Aphidiinae wasps collected in the plants did not differ between treatments, neither 6 days after the first application (F_4,8_ = 2.66, *p* = 0.111), nor 6 days after the second application (F_4,8_ = 2.12, *p* = 0.169). However, 69 days after the second application, significant differences were found (F_4,8_ = 4.07, *p* = 0.043), with zero specimens of Aphidiinae wasps (neither larvae in the mummified aphids nor adults) collected in the plots treated with imidacloprid.

The predatory heteropterans were scarce 6 days after the first and the second applications (with only seven specimens recorded over the different plots) and data recorded in these samplings were not subjected to ANOVA. On day 69 after the second application, the predatory true bugs became relatively more abundant, and significant differences between treatments and the control (F_4,8_ = 5.48, *p* = 0.020) were found, with the lowest numbers of heteropterans predators recorded in the imidacloprid and cypermethrin treatments, whereas values for the untreated control and the teflubenzuron and emamectin benzoate plots were similar.

No syrphid larvae were observed in the plots before the first application. Whereas six days after the first application, numbers of syrphid larvae did not differ among treatments (F_4,8_ = 3.09, *p* = 0.082), significant differences were found 6 days after second application (F_4,8_ = 5.39, *p* = 0.021), with zero specimens collected in the imidacloprid, cypermethrin and teflubenzuron treatments.

Since chrysopid larvae appeared at the later stages of the crop phenology, short-term effects of the insecticides on their numbers could not be observed. However, 69 days after the second application, significant differences in numbers of chrysopid larvae between the insecticide treatments and the untreated control were observed (F_4,8_ = 4.35, *p* = 0.037), with zero specimens collected in the imidacloprid plots, and very low numbers as well in the cypermethrin and emamectin benzoate plots. Incidence of chrysopid larvae in the teflubenzuron treatment was similar to that in the control.

The natural enemies were also examined in the pitfall traps. The most abundant morphospecies collected from 24 September 2017 to 11 December 2017 were the arachnid *Laminacauda* sp., the ground beetle *Blennidus peruvianus* Dejean and the wasp *Trimorus* sp.; the mean cumulative number of these morphotypes was compared between treatments and control (Table 3). There were no significant differences between treatments in numbers of *Laminacauda* sp. (F_4,8_ = 2.21, *p* = 0.159), *B. peruvianus* (F_4,8_ = 0.61, *p* = 0.669) and *Trimorus* sp. (F_4,8_ = 0.78, *p* = 0.568).

When examining the natural enemies collected in the pan traps (recorded from 24 September 2017 to 11 December 2017), three groups were the most abundant: Dolichopodidae, Syrphidae and Aphidiinae. There were no significant differences between treatments in the numbers of adults of Dolichopodidae (F_4,8_ = 1.14, *p* = 0.403) and Syrphidae (F_4,8_ = 3.33, *p* = 0.069). The number of Aphidiinae wasps was similar when comparing each treatment with the untreated control, but in the imidacloprid plot significantly lower numbers were found than in the teflubenzuron and emamectin benzoate treatments (F_4,8_ = 5.33, *p* = 0.022).

## 3. Discussion

The side effects of insecticides applied in staple crops (i.e., vegetables, legumes, rice, maize, citrus) and industrial crops (i.e., cotton, sugarcane, sugar beet) have been widely studied [16]. Thus, relevant knowledge has been gained to improve integrated pest management schemes, taking the biological control services offered by a biodiverse agroecosystem into consideration [30]. Hitherto, however, little is known about the unintentional effects of insecticides used in quinoa on non-target organisms [8]. The present field study provides information about the effects of four insecticides from different chemical groups (teflubenzuron, emamectin benzoate, imidacloprid and cypermethrin) on target and non-target arthropods in quinoa, assessed with three sampling methodologies: pitfall trapping for the ground dwelling arthropods, plant sampling for those that dwell on the quinoa plants and pan trapping for the insects that fly just above the crop canopy. 

When an insecticide is incorporated into the cropping system, changes in the structure, richness and composition of the plant dwelling arthropod community may occur, which may eventually lead to a disruption of the ecosystem services provided by the beneficial fauna [37,38,39,40]. In the present study, foliar application with imidacloprid appeared to have a higher impact on arthropods residing in the quinoa crop than the other insecticides. For example, the richness in plant dwelling species was significantly lower in the imidacloprid treatment and also the species composition differed significantly from that in the other treatments and the untreated control.

Given that the soil surface of an agricultural system is in permanent interaction with the higher strata, changes in the plant dwelling arthropod community tend to precede changes in the structure of the ground dwelling species community [23]. In this context, the reduction in species richness and the change in composition of species residing on the quinoa plants in the plots treated with imidacloprid may be related to the lower values of Shannon and Margalef indices found at the ground level as compared to the other treatments and the untreated control. The changes in the plant-associated community as a consequence of the insecticide application may have broken food webs that affected the incidence of a variety of species [41,42,43]. This may eventually be reflected in an altered species evenness and a reduction of species richness at the soil surface level, as observed in the present study [44]. On the contrary, no differences between treatments and control were found in terms of species composition and diversity of the insects collected in the yellow pan traps, probably due to the greater interaction at the top of the crop canopy with the areas surrounding the plots.

All of the tested insecticides substantially reduced densities of *S. recurvalis* larvae after the first application, which is in line with several previous studies that have demonstrated their efficacy against lepidopteran larvae [29,45,46,47,48,49,50]. As re-infestation by *S. recurvalis* larvae did not occur in any of the plots, including the control, long-term effects of the insecticide treatments to control this pest could not be judged. 

Whereas the efficacy of imidacloprid to control lepidopteran larvae by direct contact action has been demonstrated, this insecticide is also known to have an excellent systemic activity and therefore, the target organisms are mainly sucking insects such as thrips, aphids, whiteflies, leafhoppers and true bugs [29]. Accordingly, the population densities of *M. euphorbiae*, *F, occidentalis* and *N. simulans,* which recurrently infested our plots, were significantly affected by the imidacloprid treatment. 

The short-term effect of imidacloprid on *M. euphorbiae* was similar to that of the cypermethrin treatment. Differences between both treatments could be noted 69 days after the second application, with the imidacloprid plot having the lowest number of aphids per plant, likely due to its widely documented residual effects [23,29,51,52,53,54]. On the other hand, teflubenzuron and emamectin benzoate had lesser effects on the aphids as compared to imidacloprid and cypermethrin; their impact on the aphid population as compared to the untreated control was noted 6 days after the first application. Teflubenzuron is reported to have low contact activity, but due to its systemic action in the plant it may cause toxicity to aphids by ingestion [29,32,55,56]. Contrarily, emamectin benzoate has no systemic activity but can kill the exposed aphids by direct contact [29,57].

Imidacloprid and cypermethrin had similar effects on *F. occidentalis* numbers, 6 days after both the first and second application. However, the residual effect of imidacloprid appeared to have prevented the infestation to a higher degree 69 days after the second application, resulting in the lowest number of thrips per plant. The effects of teflubenzuron and emamectin benzoate treatments on *F. occidentalis* were observed 6 days after the second application. On day 69 after the second treatment the residual activity of teflubenzuron may explain the significantly lower mean number of thrips per plant (26.3 specimens) as compared to the emamectin benzoate treatment.

No visible short-term effects were observed on the population density of *N. simulans,* in any of the treatments, probably because the infestation was very low at the early stages of the crop when the insecticide treatments were done, and also because this pest was more abundant on the soil at this time. However, by its residual effect imidacloprid may have prevented a higher level of infestation by *N. simulans*, resulting in the lowest number of individuals per plant 69 days after the second application.

Given its broad-spectrum activity, imidacloprid may also affect non-target arthropods [29]. These non-target organisms may be exposed to imidacloprid by direct contact, but the compound being highly systemic, non-target omnivorous insects (including natural enemies) that feed on plant fluids or pollen may also be exposed, even a relatively long time after an application [58]. Furthermore, reduction of prey densities (i.e., the target organisms) may eventually affect the beneficial fauna that will not be able to find sufficient food, facing a greater intra- and interspecific competition [59]. In this context, with *M. euphorbiae* being highly affected by the imidacloprid treatment, there was a tendency towards lower numbers of individuals (even zero) of the aphidophagous guild on the plants as compared to the untreated control, both for the specialized natural enemies (such as the Aphidiinae wasps and predatory Syrphidae larvae) as for generalist predators (such as predaceous true bugs and chrysopid larvae). These observations are in line with previous studies indicating that imidacloprid affects key natural enemies of aphids such as coccinellids, anthocorids, geocorids, chrysopids, and Aphidiinae wasps [22,23,53,60,61]. 

Due to its broad-spectrum activity, cypermethrin is considered to be harmful to a range of natural enemies by direct contact, but its relatively short residual activity suggests that a recolonisation of the crop by these organisms may occur sometime after an application [29]. However, the observations in the present study indicate similar long-term effects of cypermethrin to those of imidacloprid on the different natural enemies collected. Teflubenzuron and emamectin benzoate sprays, however, tended to have less harmful effects on the beneficial fauna than the broad-spectrum insecticides used, which is in line with the literature [29,32]; although environmental risks of emamectin benzoate such as toxicity to certain non-target arthropods and aquatic organisms have been reported [33].

One limitation of this study is that the field experiment was done over a single growing season only. To demonstrate the efficacy of an insecticide, it is recommended that trials be carried out in different locations or growing seasons [62]. On the other hand, data provided about the side effects of the insecticides on the non-target species, particularly natural enemies, are in line with standard methods [63], so this information is relevant for quinoa growers in order to take actions to improve their current use of the pesticides. Moreover, the data are in line with those of previous studies [5,11,64] and the crop management practices are representative for quinoa cultivation in Peru and neighbouring countries [2,8,65].

## 4. Materials and Methods

### 4.1. Location 

The study was carried out in experimental fields belonging to the National Agrarian University La Molina in Lima, Peru (coordinates: 12°04′57.0″S, 76°56′49′W; altitude: 244 m). 

### 4.2. Experimental Units

The field trial was conducted under a stratified–randomized design with three replications. Each experimental unit consisted of one plot of 21 m^2^ (7 ridges of 0.75 cm width, 4 m length), with a plant density of 36 quinoa plants per linear meter (variety “Negra Collana”) after seedling removal. Each plot (as shown in Figure 3) was surrounded by polypropylene films (0.5 mm thickness, 1.5 m height, black colour) three days before the treatment (15 September 2017) and maintained until the time of harvest. At the beginning of the grain filling stage (on 5 November 2017), the whole experiment was covered with anti-bird netting to protect the crop from bird damage. Growing specifications of the field site are described in Table 5. 

### 4.3. Insecticide Treatments

Treatments were done with four insecticides (as formulated materials): teflubenzuron (150 g/L), emamectin benzoate (50 g/kg), imidacloprid (350 g/L) and cypermethrin (250 g/L). Water was used as a negative control. Specifications of the insecticides are described in Table 6.

The insecticides were applied using a manual sprayer (SOLO461: pressure 3 bars, capacity 5 L) with a full cone nozzle (TeeJet^®^). The sprayer was calibrated to apply 0.65 L per plot (corresponding to 300 L per ha). Before insecticide dissolution, water was acidified to a pH range of 5.0 to 6.0 (as recommended on the labels), with an acidifying product (SUPER ACID, 43% of organic and inorganic acids) at 0.05%. Similarly, water was acidified for the negative control.

Two applications at the maximum recommended field rate (Table 6) were made at flowering stage on 18 September 2017 (61 days after sowing) and 3 October 2017 (76 days after sowing). At this crop phase, the plants had reached their maximum height (ca., 1.2 m).

### 4.4. Sampling Methodology

Three sampling techniques were used for studying the arthropod fauna (insects and arachnids): pitfall trapping, for ground dwelling species; plant sampling, for foliage dwelling species (phytophagous insects and natural enemies); and, pan traps placed at the level of the top canopy (1.2 m), for flying insects. 

#### 4.4.1. Pitfall Trapping

One pitfall trap (as an experimental unit) was installed in the middle of each experimental plot six days after the first insecticide application and maintained until one day before harvest (from 24 September 2017 to 11 December 2017). Traps consisted of a polypropylene container (transparent, Ø 10 cm at opening and at bottom, 12 cm deep) with a mix of water and 40% v/v formaldehyde (9:1), and a few drops of detergent. The pitfall trap content was periodically collected (a total of 5 times) in airtight recipients (of the same dimensions as the traps) and carefully labelled to be transported to the laboratory for further processing. Thereafter, the collection fluid was replaced. 

#### 4.4.2. Plant Sampling

At each experimental plot, four samplings were performed, i.e., one day before the first application (17 September 2017), 6 days after the first application (24 September 2017), 6 days after the second application (9 October 2017) and 69 days after the second application (11 December 2017). Sampling consisted of taking three plants from crop rows 3 and 6 (Figure 3) in the 1st and 3rd samplings, and from rows 2 and 5 in the 2nd and 4th samplings. Plants near the borders of the plots were always avoided. 

In each plot, every plant was randomly selected and carefully collected: the plant (after cutting it at its base with scissors) was shaken over a container (width 26 cm × large 36 cm × height 20 cm) with a mix of water and 96% *v*/*v* ethanol (3:1), and some drops of liquid detergent. Thereafter, the sampled plants were carefully chopped into small pieces and the whole sample (including the liquid content) was transferred to an airtight container (volume 3l).

#### 4.4.3. Pan Trapping

Each pan trap consisted of a yellow polypropylene container (Ø 20 cm at opening and 18 cm at bottom, 7 cm deep) with a mix of water and 40% *v*/*v* formaldehyde (9:1), and some drops of detergent. One pan trap was installed in the middle of each experimental plot, at a height of 1.2 m (the opening at the level of the top of the crop canopy), six days after the first insecticide application and maintained until one day before harvest (from 24 September 2017 to 11 December 2017). As these traps were exposed to desiccation, they were regularly inspected and when needed, they were re-filled with the same collection fluid. 

The pan trap content was periodically collected (5 times) in airtight recipients (500 ml of capacity) and carefully labelled to be transported to the laboratory for further processing. Thereafter, the collection fluid was replaced.

### 4.5. Sample Processing 

All samples were processed at the laboratories of the Museum of Entomology “Klaus Raven Büller” of the National Agrarian University La Molina, in Lima, Peru, where the collected specimens were deposited.

#### 4.5.1. Sample Washing

The recipients containing pitfall trap and pan trap samples were poured onto a 1 mm mesh sieve and carefully washed with water, removing larger material such as stones, straw or leaves. The collected specimens were transferred to a labelled plastic container (Ø 5 cm, 6 cm length) containing 75% *v*/*v* ethanol for conservation and further processing (i.e., morphotyping).

The recipient with the plant samples was decanted through a 1 mm mesh sieve and carefully washed. Then, the plant parts (leaves, stem and panicle) were examined under a binocular stereoscope (Carl Zeiss: Stemi 508) to check for the presence of mines and to collect the insects that remained stuck to the plant. These specimens and those which easily detached from the plant materials were transferred to a labelled plastic container (Ø 5 cm, 6 cm length) containing 75% *v*/*v* ethanol for conservation and morphotyping.

#### 4.5.2. Morphological Identification

The specimens were examined using a binocular stereoscope (Carl Zeiss: Stemi 508) and sorted on the basis of morphological characteristics as morphospecies [66,67]. Each new morphospecies was photographed and codified, facilitating comparison when a new similar morphospecies was found, and then placed in a glass vial (Ø 1.5 cm, 4 cm length) with 75% *v*/*v* ethanol for preservation. When necessary, the morphotypes were re-examined. Each morphospecies was counted and classified at family level with the help of taxonomic keys from the literature [68].

The most abundant morphospecies were identified to genus level and, when possible, to level species with a help of specific taxonomic keys as follows: *B. peruvianus* [69,70]; *L. hyalinus* [71]; *N. simulans* [72], *L. huidobrensis* [73,74], *Rhinacloa* sp. [75], *M. tenellus* [76,77]; *N. capsiformis* [78,79,80]; *S. recurvalis* [81]. The identification of the Araneae families and genera (i.e., *Laminacauda* sp.) was assisted by arachnologist Manuel Andía associated to the Museum of Entomology “Klaus Raven Büller”.

#### 4.5.3. Molecular Identification

DNA extraction and PCR procedures were performed at the Department of Plants and Crops of Ghent University in Belgium to identify and/or confirm the species *L. huidobrensis*, *M. euphorbiae*, *L. hyalinus* and *F. occidentalis,* following the protocols provided in the literature [82,83,84,85,86].

DNA was extracted from a single specimen (*M. euphorbiae, F. occidentalis*) or a leg (*L. hyalinus*, *L. huidobrensis*) that was removed from an adult specimen with a fine cutter. The sample was transferred to a 1.5 mL Eppendorf and then crushed with a plastic rod with 20 μL of STE-buffer (100 mM NaCl, 10 mM Tris-HCL, 1 mM EDTA, pH 8.0) and 2 μL of proteinase K (10 mg/mL). This mix was incubated at 60 ° C for 30 min. Then, the activity of the proteinase K was stopped at 95 °C for 5 min.

DNA samples were subjected to PCR analysis with the primers LCO1490 FW and HCO2198RV (for *L. hyalinus* and *L. huidobrensis*); MTD 7.2 F and COI-MTD 9.2 R (for *F. occidentalis*); and, C1-J-1718 and C1-N-2191 (for *M. euphorbiae*). Amplification was performed in 50 μL total mix reaction, containing 2 μL of DNA sample, 0.25 μL GoTaq^®^ DNA Polymerase (5 u/μL), 3 μL MgCl2 solution (25 mM), 1 μL dNTPs (10 μM each), 2.5 μL forward primer, 2.5 μL reverse primer, 10 μL: 5× Colorless GoTaq^®^ Flexi Buffer, 28.75 μL water. This solution was placed in a thermal cycler with the following parameters: 2 min at 95 °C, 35 cycles of 30 s at 95 °C, 30 s at 66 °C, 45 s at 72 °C, and a final extension of 5 min at 72 °C. After amplification, 10 μL of the PCR products were subjected to electrophoresis on a 2% agarose gel, and PRC products were purified using the EZNA^®^ Cycle Pure Kit (Omega Bio-Tek) following the manufacturer’s protocols. Bidirectional Sanger sequencing, using the PCR primers, was outsourced to LGC genomics (Germany).

### 4.6. Statistical Analysis

For each sampling methodology applied, differences in the collected species composition between treatments were evaluated with the Bray–Curtis dissimilarity index and NMDS (non-metric multidimensional scaling) using the presence and abundance of the morphospecies to detect distances between plots. Significant differences between treatments were assessed using the PerMANOVA test (999 permutations).

The effects of the insecticide treatments on the diversity of the arthropod community, per each sampling methodology, were analysed through (a) the rank abundance curves and the indices of Shannon and Simpson’s dominance to evaluate the structure of the community (evenness and dominance of species) and (b) the Margalef index to assess the species richness. They were calculated for each experimental plot.

The diversity indices, mean numbers of the major pests and mean numbers of natural enemies were compared, according to each sampling methodology, between treatments by one-way ANOVA and Duncan tests, after having tested the normality and homoscedasticity of the data through Shapiro–Wilk and Bartlett tests, respectively. When the data did not meet the assumption of homogeneity of variances, the Box–Cox transformation method was applied to stabilize the variance; however, untransformed data are presented in the tables.

All statistical analyses were performed using R software, version 3.4.2 [87]. The tests were analysed at a significance level of α = 0.05.

## 5. Conclusions

The results of this study indicate that due to the detrimental effects of imidacloprid on arthropod diversity, on the composition of species and specifically on the natural enemy population, foliar application of this active ingredient is less suitable for an IPM programme in quinoa as compared to the other insecticides, in spite of its good performance in the control of the target pests. Teflubenzuron and emamectin benzoate substantially suppressed *S. recurvalis* larvae, with less negative effects than imidacloprid and cypermethrin to the beneficial fauna; however, further research is needed to evaluate the efficacy of both selective insecticides against quinoa pests in order to be considered as an element of an IPM package in quinoa. Due to the negative impact of cypermethrin on the natural enemy complex, restricted use is recommended for the management of quinoa pests.

## Figures and Tables

**Figure 1 plants-10-01788-f001:**
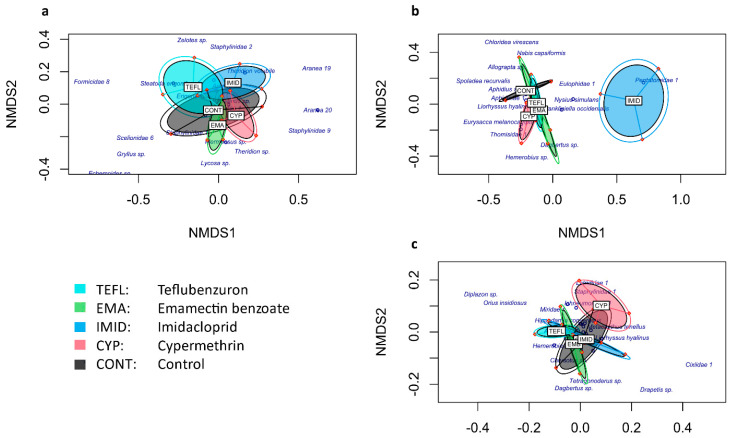
The NMDS plot showing the compositional distance between treatments for different sampling methods: (**a**). pitfall trapping; (**b**). plant sampling; (**c**). pan trapping. Plots are displayed by orange dots; plots that belong to the same treatment are fitted in a single ellipse. Each treatment is represented by a different colour.

**Figure 2 plants-10-01788-f002:**
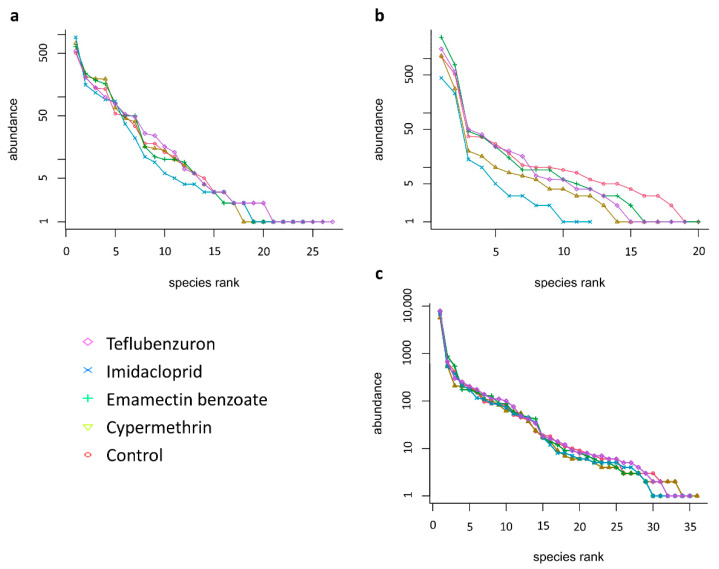
Rank abundance curves for the morphospecies found with the different sampling methodologies, per treatment (log series distribution): (**a**). pitfall trapping; (**b**). plant sampling; (**c**). pan trapping.

**Figure 3 plants-10-01788-f003:**
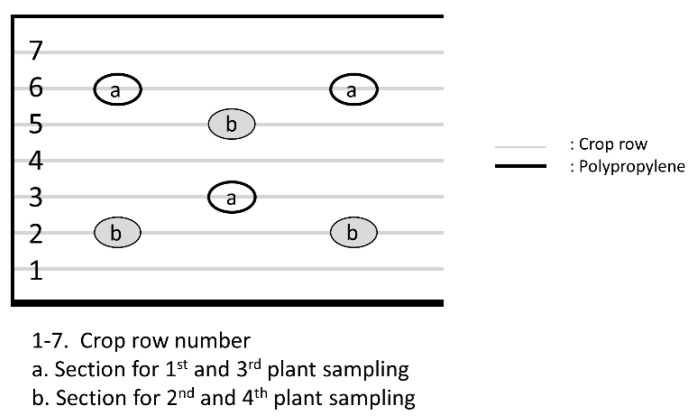
Schematic diagram of plant sampling setup for an experimental plot.

**Table 1 plants-10-01788-t001:** Diversity index (mean ± standard deviation) of morphospecies, according to the sampling methodology applied, collected after the first insecticide treatment (sampling period between 24 September 2017 and 11 December 2017).

Diversity Index	Treatments					F Value	*p*-Value
	Teflubenzuron	Emamectin Benzoate	Imidacloprid	Cypermethrin	Control		
Pitfall trapping							
Shannon	1.92 ± 0.11 ^a^	1.80 ± 0.01 ^a^	1.44 ± 0.11 ^b^	1.73 ± 0.17 ^a^	1.87 ± 0.25 ^a^	4.11	0.042
Simpson’s dominance	0.77 ± 0.03 ^a^	0.75 ± 0.05 ^a^	0.60 ± 0.05 ^b^	0.73 ± 0.09 ^a^	0.76 ± 0.09 ^a^	4.04	0.038
Margalef	3.06 ± 0.16 ^a^	2.85 ± 0.09 ^ab^	2.48 ± 0.12 ^c^	2.68 ± 0.21 ^bc^	2.78 ± 0.16 ^ab^	5.55	0.019
Plant sampling							
Shannon	0.99 ± 0.08 ^a^	0.84 ± 0.15 ^a^	0.94 ± 0.18 ^a^	0.82 ± 0.13 ^a^	1.13 ± 0.06 ^a^	2.57	0.119
Simpson’s dominance	0.49 ± 0.06 ^a^	0.43 ± 0.09 ^a^	0.52 ± 0.10 ^a^	0.40 ± 0.08 ^a^	0.55 ± 0.02 ^a^	1.81	0.220
Margalef *	1.71 ± 0.13 ^a^	1.71 ± 0.26 ^a^	1.09 ± 0.27 ^b^	1.78 ± 0.06 ^a^	1.97 ± 0.52 ^a^	4.76	0.029
Pan trapping							
Shannon	1.11 ± 0.14	1.13 ± 0.19	1.09 ± 0.01	1.18 ± 0.10	1.13 ± 0.05	0.34	0.841
Simpson’s dominance	0.40 ± 0.06	0.43 ± 0.08	0.04 ± 0.01	0.44 ± 0.04	0.42 ± 0.04	0.45	0.771
Margalef *	3.52 ± 0.29	3.40 ± 0.08	3.50 ± 0.20	3.61 ± 0.15	3.61 ± 0.23	0.49	0.747

Different letters within a row indicate significant differences at α = 0.05 (Duncan test), when the ANOVA was significant. * ANOVA run after using Box–Cox transformation method (γ = 0.2).

**Table 2 plants-10-01788-t002:** Numbers of individuals of the major insect pests (mean no. per plant ± standard deviation) under different treatments.

Taxa	Treatments					F Value	*p*-Value
	Teflubenzuron	Emamectin Benzoate	Imidacloprid	Cypermethrin	Control		
*Spoladea recurvalis*							
1DBA	2.4 ± 1.39	2.8 ± 0.51	2.7 ± 0.33	3.2 ± 1.64	2.4 ± 1.02	0.15	0.956
1st application							
* 6DAA	0.4 ± 0.77 ^b^	0.1 ± 0.19 ^b^	0 ± 0.0 ^b^	0.1 ± 0.19 ^b^	2.0 ± 0.88 ^a^	7.73	0.007
2nd application							
6DAA	0	0	0	0	0.67 ± 0.67	N.A.	N.A.
69DAA	0	0	0	0	0	N.A.	N.A.
*Macrosiphum euphorbiae*							
1DBA	11.4 ± 4.33	15.7 ± 9.17	13.1 ± 4.19	20.4 ± 10.49	12.3 ± 4.26	1.33	0.338
1st application							
** 6DAA	11.6 ± 4.74 ^b^	7.8 ± 0.77 ^b^	1.8 ± 2.04 ^c^	2.9 ± 2.99 ^c^	27.4 ± 10.83 ^a^	28.73	<0.001
2nd application							
6DAA	6.9 ± 4.44 ^a^	11.2 ± 3.56 ^a^	0.2 ± 0.19 ^b^	0.3 ± 0.33 ^b^	6.6 ± 3.89 ^a^	7.32	0.008
*** 69DAA	145 ± 40.19 ^ab^	250 ± 104.46 ^a^	36.2 ± 1.67 ^c^	113.8 ± 25.06 ^b^	86.1 ± 27.48 ^bc^	7.80	0.007
*Frankliniella occidentalis*							
1DBA	2.2 ± 1.26	2.4 ± 1.26	2.9 ± 0.84	4.3 ± 0.58	1.8 ± 0.68	2.62	0.115
1st application							
6DAA	5.3 ± 1.20 ^a^	4.7 ± 1.15 ^a^	2.3 ± 0.33 ^b^	1.4 ± 0.38 ^b^	5.4 ± 1.17 ^a^	9.47	0.004
2nd application							
*** 6DAA	5.1 ± 0.84 ^b^	4.3 ± 1.15 ^b^	2.1 ± 0.51 ^c^	0.6 ± 0.19 ^c^	9.89 ± 2.46 ^a^	171.17	<0.001
69DAA	26.3 ± 11.1^b^	62.1 ± 6.50 ^a^	7.2 ± 1.89 ^d^	15.1 ± 5.42 ^bc^	20.3 ± 11.98 ^bc^	46.76	<0.001
*Nysius simulans*							
* 1DBA	0.1 ± 0.19	0.2 ± 0.19	0.1 ± 0.19	0.3 ± 0.33	0.8 ± 0.84	0.82	0.549
1st application							
6DAA	0 ± 0	0 ± 0	0.2 ± 0.19	0.1 ± 0.19	0.1 ± 0.19	1.75	0.232
2nd application							
* 6DAA	0.6 ± 0.38	0.6 ± 0.69	0.3 ± 0.33	0.1 ± 0.19	0 ± 0	1.62	0.261
69DAA	5.0 ± 0.67 ^a^	3.6 ± 0.19 ^b^	1.0 ± 0.67 ^d^	2.0 ± 0.33 ^c^	2.9 ± 0.19 ^bc^	25.87	<0.001

DBA: days before application; DAA: days after application; N.A.: not applicable. Different letters within a row indicate significant differences at α = 0.05 (Duncan test), when the ANOVA was significant. * ANOVA run after using Box–Cox transformation method γ =−2.0, ** γ = 0.45, *** γ = 0.30.

**Table 3 plants-10-01788-t003:** Cumulative numbers (mean no. per trap ± standard deviation) of the most abundant phytophagous insects and natural enemies, collected with two sampling methodologies, after the second insecticide application (sampling period from 24 September 2017 to 11 December 2017).

Taxa	Treatments					F Value	*p*-Value
	Teflubenzuron	Emamectin Benzoate	Imidacloprid	Cypermethrin	Control		
Pitfall trapping							
*Nysius simulans*	181.3 ± 82.6	214.0 ± 83.5	301.3 ± 49.9	241.0 ± 113.9	169.0 ± 89.9	1.62	0.261
*Laminacauda* sp.	80.0 ± 30.3	115.3 ± 9.9	67.3 ± 11.8	135.7 ± 55.4	90.7 ± 26.7	2.21	0.159
*Blennidus peruvianus*	68.7 ± 28.0	79.7 ± 38.4	52.0 ± 3.46	65.3 ± 30.6	68.3 ± 4.6	0.61	0.669
*Trimorus* sp.	33.7 ± 16.3	33.3 ± 7.5	19.7 ± 4.7	35.7 ± 17.9	28.0 ± 10.6	0.78	0.568
Pan traps							
*Nysius simulans*	36.7 ± 4.73	29.0 ± 10.5	38.7 ± 12.3	36.3 ± 10.7	32.3 ± 11.6	0.42	0.792
*Frankliniella occidentalis*	2634.0 ± 188.9 ^a^	2559.3 ± 84.1 ^ab^	2276.3 ± 205.0 ^b^	1892.0 ± 54.7 ^c^	2311.0 ± 59.6 ^b^	11.18	0.002
Dolichopodidae	100.0 ± 38.2	179.7 ± 121.3	124.3 ± 34.7	53.3 ± 42.3	138.7 ± 125.7	1.14	0.403
Syrphidae	25.3 ± 2.1	19.3 ± 4.7	11.7 ± 3.8	18.3 ± 8.4	15.0 ± 2.64	3.33	0.069
Aphidiinae *	33.3 ± 13.0 ^ab^	51.0 ± 25.2 ^a^	18.0 ± 8.7 ^c^	20.7 ± 5.5 ^bc^	27.0 ± 7.0 ^abc^	5.33	0.022

Different letters within a row indicate significant differences at α = 0.05 (Duncan test), when the ANOVA was significant. * ANOVA run after using Box–Cox transformation method γ = −0.4.

**Table 4 plants-10-01788-t004:** Numbers of individuals of the most abundant insect natural enemies collected (mean no. per plant ± standard deviation) under different treatments.

Taxa	Treatments					F Value	*p*-Value
	Teflubenzuron	Emamectin Benzoate	Imidacloprid	Cypermethrin	Control		
**Aphidiinae wasps**							
1DBA	1.66 ± 1.53	1.67 ± 0.88	2.22 ± 1.35	1.44 ± 0.84	1.00 ± 0.67	0.96	0.481
1st application							
* 6DAA	0.22 ± 0.19	0.56 ± 0.19	0.11 ± 0.19	0.22 ± 0.39	3.44 ± 4.28	2.66	0.111
2nd application							
* 6DAA	0.33 ± 0.0	0.55 ± 0.69	0.0 ± 0.0	0.11 ± 0.19	0.11 ± 0.19	2.12	0.169
** 69DAA	0.22 ± 0.19 ^ab^	0.11 ± 0.19 ^b^	0.0 ± 0.0 ^b^	0.22 ± 0.19 ^ab^	0.56 ± 0.20 ^a^	4.07	0.043
**Predatory true bugs**							
1DBA	0.0 ± 0.0	0.2 ± 0.17	0.23 ± 0.40	0.33 ± 0.35	0.0 ± 0.0	N.A.	N.A.
1st application							
* 6DAA	0.10 ± 0.10	0.03 ± 0.06	0.03 ± 0.06	0.0 ± 0.0	0.0 ± 0.0	N.A.	N.A.
2nd application							
6DAA	0.22 ± 0.39	0.0 ± 0.0	0.0 ± 0.0	0.0 ± 0.0	0.22 ± 0.19	N.A.	N.A.
69DAA	1.67 ± 0.67 ^a^	1.78 ± 1.01 ^a^	0.11 ± 0.19 ^b^	0.22 ± 0.39 ^b^	1.89 ± 0.69 ^a^	5.48	0.020
**Syrphid larvae**							
1DBA	0.0	0.0	0.0	0.0	0.0	N.A.	N.A.
1st application							
* 6DAA	0.23 ± 0.40	0.33 ± 0.58	0.0 ± 0.0	0.10 ± 0.17	1.57 ± 1.25	3.09	0.082
2nd application							
* 6DAA	0.0 ± 0.0 ^b^	0.67 ± 1.15 ^ab^	0.0 ± 0.0 ^b^	0.0 ± 0.0 ^b^	0.56 ± 0.19 ^a^	5.39	0.021
1DBA	0.0	0.0	0.0	0.0	0.0	N.A.	N.A.
**Chrysopid larvae**							
1DBA	0.0	0.0	0.0	0.0	0.0	N.A.	N.A.
2nd application							
6DAA	0.0	0.0	0.0	0.0	0.0	N.A.	N.A.
69DAA	1.0 ± 0.0 ^ab^	0.53 ± 0.69 ^bc^	0.0 ± 0.0 ^c^	0.56 ± 0.51 ^bc^	1.56 ± 0.51 ^a^	4.35	0.037

DBA: days before application; DAA: days after application; NA: not applicable. Different letters within a row indicate significant differences at α = 0.05 (Duncan test), when the ANOVA was significant. * ANOVA run after using Box–Cox transformation method γ = −2.5; ** γ = 0.3.

**Table 5 plants-10-01788-t005:** Growing specifications of the experimental field.

	Specifications	Dates
Sowing	drilling sowing method	19 July 2017
Harvest	threshing	12 December 2017
Irrigation	surface irrigation	(20 July 2017; 17 August 2017; 7 September 2017; 28 September 2017; 25 October 2017; 9 November 2017)
Fertilisation doses (NKP)	160–80–160	19 July 2017
Soil type	clay loam	
Neighbouring crops	Quinoa (*Chenopodium quinoa*), Wheat (*Triticum* spp.) Corn (*Zea mays*) Kiwicha (*Amaranthus caudatus*)	
Fungicides	1° benzomyl 2° metalaxyl + mancozeb 3° dimetomorph	(24 July 2017) (7 August 2017) (25 August 2017)
Weed management	Manual control	(25 July 2017, 11 August 2017; 19 November 2017)
Previous crop	Fallow period of 4 months	

**Table 6 plants-10-01788-t006:** Insecticide specifications used in the treatments.

Insecticide	Label Field Rate (g a.i. ha^−1^) *	Chemical Group	Commercial Name	Company
Cypermethrin	75	Pyrethroid	Cypmor 25 EC	Jebsen and Jessen Peru S.A.C.
Teflubenzuron	33.75	Benzoylphenylurea	Mercury 150 SG	Point Andina S.A.
Emamectin benzoate	10	Avermectin	Olimpo 5% SG	Sharda Peru S.A.C.
Imidacloprid	131.25	Neonicotinoid	Phantom	Jebsen and Jessen Peru S.A.C.

a.i. = active ingredient. * spray liquid applied at a rate of 300 L/ha.

## Data Availability

The data presented in this study are available upon request from the corresponding author.

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
