# Peer review of "Field Evaluation of Cypermethrin, Imidacloprid, Teflubenzuron and Emamectin Benzoate against Pests of Quinoa (Chenopodium quinoa Willd.) and Their Side Effects on Non-Target Species"

_plants, 2021, doi:10.3390/plants10091788_

Round 1

Reviewer 1 Report

The authors studied the effects of two broad-spectrum insecticides
(cypermethrin, imidacloprid) and two selective insecticides (flubenzuron and emamectin benzoate), with different mechanism of action and from different chemical groups, against quinoa pests in Peru and recorded their side effects on non-target arthropods by species composition, species diversity and  dynamics of population in quinoa fields at coastal level.
Although it is a relatively well-crafted manuscript that yields
interesting results, the MS cannot be recommended for publication because it does not meet the high standards set by scientific journals for this type of biological test.
Main shortcomings:
1. Biological tests were performed at only one locality and during one season. Based on the data thus obtained, safety or environmental risks cannot be assessed. Therefore, only biological effect against target organisms can be assessed (and still to a limited extent). For comprehensive conclusions, it is necessary to obtain relevant data from several different localities or from several years.
2. Due to the mechanism of action, the efficacy should also be evaluated after 24-48 hours after application.
3. Their recommended conclusion contradicts some studies. The authors recommend teflubenzuron and emamectin benzoate as safe active ingredients that are suitable for IPM. According to EFSA (EFSA Journal 2012; 10 (11): 2955), for example, Emamectin is highly toxic to non-target aquatic organisms, arthropods and bees. However, this fact is not properly discussed by the authors.

Reviewer 2 Report

The reviewer appreciate the research work done by the authors, the manuscript is well written, structured and the data presented clearly.

Regarding the data presented by the authors, in terms of the insecticidal activity showed a toxicty comparison different insecticides to different quinoa pests and natural enemies, useful for IPM programs. For the data of composition of species, arthrophod diversity, etc. new data from different growing season it should be added to support better the conclusions. 

Although the authors indicate this limitation in the manuscript, they need to provide more refrences to prove that the data are in agreement with other studies at the time that the study have been done. 

Round 2

Reviewer 1 Report

The authors answered the reviewer's questions. I have no further questions. 

Reviewer 2 Report

The modification in the manuscript provide enough information to answer the questions of this reviewer. The recomendation of this reviewer is accept the article to be published by the journal.